# Evaluation of Recombinant Bovine Interleukin-8 (rbIL-8) as a Treatment for Chronic Intramammary Infection in Dairy Cows

**DOI:** 10.3390/antibiotics11081029

**Published:** 2022-07-30

**Authors:** Phillip M. G. Peixoto, Lais L. Cunha, Leonardo Barbosa, Wilson Coelho, Giorgia Podico, Rodrigo C. Bicalho, Igor F. Canisso, Fabio S. Lima

**Affiliations:** 1Department of Veterinary Clinical Medicine, University of Illinois, Urbana, IL 61802, USA; martinsg.phillip@ufl.edu (P.M.G.P.); laiscvet@gmail.com (L.L.C.); leobarbosar95@gmail.com (L.B.); wmcoelhojr@ucdavis.edu (W.C.J.); gpodico@illinois.edu (G.P.); canisso@illinois.edu (I.F.C.); 2Department of Large Animal Clinical Sciences, D. H. Barron Reproductive, and Perinatal Biology Research Program, University of Florida, Gainesville, FL 32610, USA; 3Department of Population Health and Reproduction, School of Veterinary Medicine, University of California, Davis, CA 95616, USA; 4Department of Comparative Biosciences, University of Illinois, Urbana, IL 61802, USA; 5Fera Diagnostics and Biologicals Corporation, College Station, TX 77802, USA; bicalho@feraah.com

**Keywords:** subclinical mastitis, SCC, interleukin 8

## Abstract

Mastitis is one of the main contributors to antimicrobial resistance in livestock, so alternative therapies are being investigated to address it. The present study assessed the capability of recombinant bovine interleukin-8 (rbIL-8) to improve neutrophil function in the mammary gland and resolve chronic high somatic cell count (SCC) in Holstein cows. Multiparous cows (n = 8) with more than 300,000 SCC per mL were allocated to one of two intramammary infusions: saline (10 mL of saline solution) or rbIL-8 (1.57 mg/mL of recombinant bovine IL-8 diluted in 9 mL of saline). In addition, there was an untreated control group (n = 2, SCC < 300,000 SCC/mL). Milk samples were collected post-treatment at 0, 4, 8, 12, 24, 48, and 144 h to quantify milk SCC, haptoglobin, and IgG concentrations. Neutrophil’s phagocytosis in milk and blood was evaluated via flow cytometry at 0, 24, and 48 h. The log of SCC did not differ between the infused groups (*p* = 0.369). Neutrophils presented a similar log of cells with high fluorescence for propidium-iodide (PI) and dihydrorhodamine (DHR) in milk (*p* = 0.412) and blood samples (*p* = 0.766) in both infused groups. Intramammary infusion of 1.57 mg/mL of rbIL-8 did not improve neutrophils response and failed to resolve chronic high SCC.

## 1. Introduction

Mastitis remains the most prevalent and costly disease in dairy cattle in the United States [1]. The main therapeutic choice farmers adopt to address this disease is using antibiotics [1]. Conversely, about 70% of the annual intramammary expenses designated for mastitis cases do not show benefits from antimicrobial usage [2]. Most of these costs are associated with cases of spontaneous cure and treated cases that resulted in clinical cure failure [3,4,5]. Cure failure is mainly attributed to antimicrobial resistance and the fact that some cases are not caused by bacteria, considering that between 25 and 40% of cultured mastitis cases yield a negative result [2,4,6].

Approximately 48% of costs related to mastitis treatments are attributed to its subclinical form [7]. In general, subclinical mastitis presents as a response to an insult that triggers neutrophil migration to the mammary gland, characterized by an increase in somatic cell count to >200,000 cells/mL, without visible udder inflammation [8]. In addition, subclinical mastitis is associated with decreased milk production ranging from 0.77 to 1.78 kg/day in multiparous Holstein cows [9] and increased culling risk [10]. Another essential feature of this mastitis form is the lack of resolution in some cases, leading to chronic high SCC, a condition marked by a persistent elevated SCC for over two or more monthly milk tests [11]. Chronic high SCC is associated with a more profound milk production loss, ranging from 4.1 to 5.7 kg/day compared to healthy cows, whereas the non-chronic subclinical form loses 2.1 kg/day [12]. The progression of this disease into chronic mastitis, especially in *Staphylococcus aureus*-derived cases, is biologically associated with impaired neutrophil function and pathogens’ ability to subvert the host immune response locally in the mammary gland [13]. Therefore, interventions that can revamp the neutrophil population and functionality in the mammary gland might be a surrogate measure to treat chronic infections. Interleukin-8 is one of the main chemoattractants for neutrophils. The binding of IL-8 to its receptors (CXCR1 and CXCR2) induces neutrophil activation, stimulates chemotaxis, and increases phagocytosis and killing ability [14,15]. Mitchel et al. (2003) [16] reported that the infusion of IL-8 into calves’ lungs, using a bronchoscope, elicited local neutrophil recruitment. Additionally, histological samples collected from the calves’ bronchioles treated with IL-8 showed that neutrophils populated the alveoli. In vitro assessment of neutrophils collected from these calves previously treated with IL-8 showed an increase in mean fluorescence related to phagocytosis and oxidative burst when compared to controls [16]. An experiment performed on cows at dry-off, receiving an intramammary treatment with recombinant IL-8, induced migration of neutrophils, increased the percentage of polymorphonuclear (PMN) cells collected from milk samples and increased the concentration of IgG2, which usually contains opsonic properties [17]. Moreover, the treatment of IL-8 decreased the concentration of casein, which inhibits myeloperoxidase (MPO)-mediated oxygen-dependent bactericidal activity of neutrophils [17]. Intravaginal and intravenous infusion of recombinant interleukin 8 (rbIL-8) elicited a transient increase in circulating and endometrial PMN, mitigating the development of puerperal metritis in dairy cows [18]. One of the concerns of using rbIL-8 in animals is its pyrogenic effect, which raises welfare issues. The infusion of IL-8 in the third ventricle (via guiding cannula) in rats led to an increase in temperature of about 0.5 °C, compared with the saline infusion [19]. Additionally, intravenous infusion of rbIL-8 in rabbits also induced a pyrogenic response [20]. In cows, the intravenous infusion of rbIL-8 also elicited an increase in temperature compared to untreated controls, but the intravaginal infusion did not cause this pyrogenic response [18].

Considering the compelling evidence for the use of exogenous rbIL-8 to stimulate an acute immune response, we conjecture that this molecule’s applicability in resolving chronic high SCC in dairy cows warrants further investigation. Our central hypothesis is that infusing an intramammary high dose of rbIL-8 promotes a steady elevation of SCC and improves neutrophil chemotaxis and killing ability in the udder, leading to a subsequent decrease in SCC and resolution of the persistent high SCC. In order to test our hypothesis, three specific aims were established: First, circulatory concentrations of PMN in the blood and the milk were quantified relative to post-intramammary treatment with rbIL-8. Second, neutrophils’ capability of phagocytosis and reactive oxidative burst was evaluated. Third, haptoglobin and IgG milk concentrations were quantified, and a blood biochemical profile was evaluated.

## 2. Results

### 2.1. White Blood Cell Count and Somatic Cell Count

The cows presenting high SCC were randomized according to the SCC and assigned to one of two intramammary treatments (Figure 1). The overall concentrations of white blood cell (WBC) count were 10.60 ± 5.94, 7.09 ± 4.20, and 16.27 ± 4.20 (mean ± SEM), respectively, for control, rbIL-8, and saline groups. Since the saline group initiated the study with greater concentrations of WBC compared to the other two groups, the data were evaluated based on the percentage change in WBC relative to the previous hour (0 to 4 h, 4 to 8 h, 8 to 24 h, and 24 to 48 h). Multiple comparison analyses revealed no statistical differences among group, hour, or group-by-hour interactions, as in the contrast analysis (Table 1). The mean ± SEM concentration of the log of SCC in the milk samples was: 4.71 ± 0.15, 6.44 ± 0.11, and 6.29 ± 0.11, respectively, for the control, rbIL-8, and saline groups. Multiple comparison analysis revealed a statistical difference between rbIL-8 vs. control (*p* < 0.001), and saline vs. control (*p* = 0.0003), but not between rbIL-8 vs. saline (*p* = 0.621). A statistical significance was recorded regarding the interaction between groups and hour (*p* < 0.001). Upon contrast analysis, the two infused groups had similar concentrations of SCC between them (*p* = 0.369), whereas the control group showed lesser concentrations of SCC compared to rbIL-8 and saline (*p* < 0.001; Table 2).

### 2.2. Evaluation of Milk and Blood Neutrophils Phagocytosis

Density plots recovered from flow cytometry evaluation are represented in Figure 2. The logarithmic amounts of milk neutrophils captured on the flow cytometry gate were 3.36 ± 0.19, 3.85 ± 0.13, and 3.89 ± 0.13 (mean ± SEM), respectively, for control, rbIL-8, and saline groups. The concentrations of neutrophils were similar among all three groups (*p* = 0.132), and there were no interactions between group and hour (*p* = 0.971). The contrast analysis showed no significant difference between infused groups regarding the neutrophil count in the milk (*p* = 0.840). The level of SCC (rbIL-8 and saline vs. control) was statistically significant (*p* = 0.052), meaning that the chronic cows had a greater count of milk neutrophils compared with the control cows: 3.87 ± 0.09 vs. 3.36 ± 0.18 (log, mean ± SEM). Neutrophils extracted from the blood samples and quantified in the flow cytometry were 3.75 ± 0.26, 3.50 ± 0.18, 3.45 ± 0.18 (log scale; mean ± SEM), respectively, for control, rbIL-8, and saline group. No statistical significance was found among groups (*p* = 0.820).

The total amount of neutrophils presenting low fluorescence for both dyes among the three groups were similar for the milk samples (*p* = 0.112) and the blood samples (*p* = 0.673). The contrast analysis for the milk neutrophils revealed no difference in the number of low-fluorescence neutrophils between the infused groups (*p* = 0.198). Still, the chronic groups tended to yield more low-fluorescence neutrophils than the control group (*p* = 0.086). When the infused groups were compared, the contrast analysis revealed no difference in low-fluorescent blood neutrophils (*p* = 0.785) or SCC level (*p* = 0.428).

Milk neutrophils that presented high fluorescence only to PI were 1.42 ± 0.37, 2.56 ± 0.26, and 2.46 ± 0.26 for control, rbIL-8, and saline, respectively (log, mean ± SEM). The multiple comparison analysis revealed that the groups tended to differ (*p* = 0.083; Figure 3A). The rbIL-8 vs control revealed a tendency (*p* = 0.085), but the saline vs. control (*p* = 0.120) and rbIL-8 vs. saline (*p* = 0.955) revealed no statistical difference. The statistical tendency reported among groups in the multiple comparison analysis is driven by the SCC level (*p* = 0.031), as opposed to infused groups (*p* = 0.779), as reported by the contrast analysis. Blood neutrophils high-fluorescence scores for PI were 1.88 ± 0.37, 1.50 ± 0.21, and 1.91 ± 0.28 for control, rbIL-8, and saline, respectively (*p* = 0.468; Figure 3B). The contrast revealed no statistical significance between the infused groups (*p* = 0.272) and the SCC level (*p* = 0.688). Milk neutrophils with high fluorescence only to dihydrorhodamine (DHR) were 1.90 ± 0.24, 2.87 ± 0.17, and 2.89 ± 0.17 for control, rbIL-8, and saline, respectively (log mean ± SE), representing a statistical difference among groups (*p* = 0.024; Figure 4A), according to the multiple comparison analysis. The rbIL-8 vs. control comparison showed a statistical significance (*p* = 0.032), as did the saline vs. control (*p* = 0.029), but the rbIL-8 vs. saline (*p* = 0.957) revealed no statistical difference. The contrast analysis revealed that rbIL-8 and saline (chronic cows) differed from the control (SCC level, *p* = 0.008). The contrast within the infused groups revealed no statistical difference (*p* = 0.946). Neutrophils presenting high fluorescence for DHR in the blood samples were 2.36 ± 0.59, 2.70 ± 0.40, and 2.57 ± 0.40 for control, rbIL-8, and saline, respectively (*p* = 0.898; Figure 4B). There was no statistical difference regarding high-fluorescence DHR-positive neutrophils between the SCC levels (*p* = 0.629) or between the infused groups (*p* = 0.832). Milk neutrophils presenting high fluorescence for DHR and PI differed among the three groups 1.14 ± 0.34, 2.92 ± 0.24, and 2.65 ± 0.24, for control, rbIL-8, and saline, respectively (*p* = 0.007; Figure 5A). Multiple comparison analyses revealed that rbIL-8 (*p* = 0.006) and saline (*p* = 0.001) differed from the control, but the rbIL-8 and saline were not different (*p* = 0.729). The contrasts analysis revealed a greater number of neutrophils that were positive for both dyes in the chronic cows compared to controls (*p* = 0.002), but not between the infused groups (*p* = 0.461). The circulatory neutrophils were 1.5 ± 0.83, 1.54 ± 0.57, and 1.33 ± 0.57, for control, rbIL-8, and saline, respectively (*p* = 0.949; Figure 5B). Both the SCC level and the infused groups had similar amounts of neutrophils with high fluorescence for DHR (*p* = 0.928) and PI (*p* = 0.766).

The mean fluorescence scores for PI in the neutrophils recovered from the milk samples were 3.83 ± 0.26, 4.63 ± 0.27, and 4.61 ± 0.27 (log, mean ± SEM) for control, rbIL8, and saline, respectively, representing a statistical tendency among them (Figure 6A, *p* = 0.070). Multiple comparison analysis revealed a tendency between rbIL-8 vs. control (*p* = 0.081), and for the saline vs. control (*p* = 0.088), but not for the rbIL-8 vs. saline (*p* = 0.997). The contrast analysis revealed a greater PI mean fluorescence for the chronic cows compared to control cows (*p* = 0.024), but no differences between the infused groups were present (*p* = 0.946). Similarly, in the circulatory neutrophils, the mean fluorescence for PI was 3.92 ± 0.31, 4.07 ± 0.22, and 4.04 ± 0.22 for control, rbIL8, and saline (log, mean ± SEM), respectively, and did not differ among the groups (Figure 7A, *p* = 0.919). The mean fluorescence scores for DHR for milk samples were 2.23 ± 0.34, 3.98 ± 0.24, and 3.74 ± 0.24 for control, rbIL8, and saline (log, mean ± S.E), respectively, and it differed among groups (Figure 6B, *p* = 0.008). Multiple comparison analysis revealed a statistical significance between rbIL-8 vs. control (*p* = 0.008), and for the saline vs. control (*p* = 0.001), but not for the rbIL-8 vs. saline (*p* = 0.785). Chronic cows had greater mean fluorescence of DHR compared with the control (*p* = 0.002). Circulatory neutrophils’ mean fluorescence for DHR was 3.5 ± 0.26, 3.75 ± 0.18, and 3.68 ± 0.18, for control, rbIL8 and saline (log, mean ± SEM), respectively, and it was similar among groups (Figure 7B, *p* = 0.757). There was an effect of hour relative to treatment in the mean fluorescence of PI and DHR for both specimens in the circulatory neutrophils (*p* < 0.05; Figure 6 and Figure 7). 

### 2.3. Evaluation of Blood Biochemical Profile, Milk Haptoglobin, and IgG 

The mean concentrations of IgG were 2641.21 ± 7.74, 2659.18 ± 5.47, and 2658.94 ± 5.47 µg/mL (mean ± S.E; *p* = 0.140), and for haptoglobin retrieved from milk samples were 32.35 ±1.27, 34.30 ± 1.18, and 33.82 ± 1.18 µg/mL (mean ± S.E; *p* = 0.983), respectively, for control, rbIL-8, and saline (Figure 8A,B). There was no significant statistical effect on SCC level (*p* = 0.854) or intramammary infusion (*p* = 0.955) regarding the haptoglobin concentrations. Still, the chronic cows differed from the control cows (*p* = 0.049), and the infused groups were similar, considering the IgG concentrations (*p* = 0.975). 

The biochemical profile was evaluated using blood samples, and it was revealed that total protein differed among groups (Table 3; *p* = 0.054). The infused groups did not differ (*p* = 0.161), but the chronic groups differed from the control (*p* = 0.003). Albumin concentrations tended to differ among groups (*p* = 0.062). The rbIL-8 differed from the control (*p* = 0.058), the saline did not differ from the control (*p* = 0.436), and the rbIL-8 did not differ from the control (*p* = 0.272) for albumin concentrations. The control group had higher concentrations of albumin compared with the chronic groups (*p* = 0.056). Globulin concentrations differed among groups (*p* = 0.032). The rbIL-8 differed from the control (*p* = 0.028), and the saline vs. control did not differ (*p* = 0.250), nor did the rbIL-8 vs. saline (*p* = 0.225), according to the multiple comparison analysis. Additionally, the chronic cows had greater concentrations of globulin compared to the control (*p* = 0.025), and the infused groups tended to differ between them (Table 3; *p* = 0.106). The infused groups were compared between them and sliced each hour (Figure 9A). The rbIL-8 presented greater concentrations of globulins at 24 h relative to treatment (*p* = 0.021). Ratio of albumin/globulin differed among groups (*p* = 0.043), and this ratio was greater in the control compared with the chronic groups (*p* = 0.036). GGT did not differ among treatments (*p* = 0.196), but the chronic groups tended to have greater concentrations of GGT compared with the control (*p* = 0.079). The concentrations of ALP had a statistical tendency regarding the treatment by hour interaction (*p* = 0.072), although SCC level (*p* = 0.887) and infused groups (*p* = 0.751) did not differ. Upon evaluation in a separate statistical model for ALP concentrations, the interaction of SCC level and hour relative to treatment showed a statistical significance (*p* = 0.036) and differed at 0 and 4 h relative to treatment (Figure 9B). Similarly, CK concentrations showed a statistical significance regarding the interaction of treatment and hour relative to treatment (*p* = 0.013), although the SCC level (*p* = 0.294) and infused groups (*p* = 0.897) did not differ. An evaluation using a separate model comparing the infused groups for CK concentrations revealed a statistical tendency of the interaction between the treatment and hour relative to treatment (*p* = 0.072). A tendency at 4 h relative to treatment between rbIL-8 and saline groups was reported (Figure 10A). A separate model was also used for the SCC level, and a statistical significance of the interaction between the category and hour relative to treatment was reported (*p* < 0.001), and the CK concentrations differed at 0, 4, and 24 h (Figure 10B). 

Circulatory concentrations of cholesterol tended to differ among groups (*p* = 0.091). The rbIL-8 tended to differ from the control (*p* = 0.098), while the saline vs. control did not differ (*p* = 0.648), nor did the rbIL-8 vs. saline (*p* = 0.235), for cholesterol concentrations in the multiple comparison analysis. The contrast analysis revealed that the control group tended to differ compared to chronic groups (*p* = 0.107), but the infused groups did not differ (*p* = 0.111). Magnesium concentrations in the blood showed a statistical significance among groups (*p* = 0.028), and on the interaction between treatment and hour relative to treatment (*p* = 0.036) in the multiple comparison analysis. The rbIL-8 differed from the control (*p* = 0.025), but the saline vs. control (*p* = 0.310) and rbIL-8 vs. saline (*p* = 0.187) did not differ, regarding the magnesium concentrations in the bloodstream. Additionally, chronic groups differed from the control groups (*p* = 0.027), and infused groups tended to differ (*p* = 0.085). A separate model evaluating only the infused groups reported a statistical significance between group and hour relative to the infusion (*p* = 0.040), and rbIL-8 decreased the magnesium concentrations compared with saline at 4 h (Figure 11).

## 3. Discussion

The objective of the current study was to evaluate the effectiveness of intramammary infusion of a recombinant interleukin 8 as a therapy to resolve chronic subclinical mastitis by decreasing the SCC and improving neutrophil phagocytic response. The intramammary infusion of saline or rbIL-8 induced an acute increase in the somatic cell count, which was similar between the two groups across hours relative to treatment. However, the SCC in the rbIL-8 presented a distinct pattern. It was observed that the SCC in the rbIL-8 group remained elevated from 12 to 48 h, whereas the other two groups declined at 24 h relative to treatment. Such behavior in the milk concentrations of somatic cells was also observed in dry-off cows receiving intramammary solution containing 25 µg of rbIL-8, in which the SCC was elevated from 24 to 720 h relative to treatment compared with the contralateral quarter infused with saline [17]. In the same investigation, the circulatory concentrations of white blood cells acutely decreased from 0 h to 72 h relative to intramammary infusion of 25 µg of rbIL-8, from 7 × 10^3^ counts/µL to about 3 × 10^3^ counts/µL [17]. The change in the circulatory concentrations of white blood cells occurred concurrently with the increase of SCC, suggesting that the circulatory PMN was migrating from the bloodstream to the quarter intramammary treated with the rbIL-8 [17]. Although the present study showed a different pattern of SCC and WBC in the rbIL-8 group, there were no significant effects between the infused groups regarding SCC and WBC. 

A biological explanation for the non-resolution of the elevated SCC in the rbIL-8 treatment is that perhaps the mammary gland of the chronic elevated SCC cows is compromised to a degree that is non-reparable by a single treatment with rbIL-8, and multiple treatments might be needed to elicit a high-magnitude response that resolves the chronic high SCC. It has been pointed out in previous reports that the quarters with subclinical mastitis (especially those cultured positive to *Staphylococcus aureus*) have a disrupted immune function, leading to non-resolution of the infection, which may cause chronic recruitment of PMN to the mammary gland [13]. In fact, in the present study, the milk culture at 144 h relative to treatment revealed no difference in the bacteriological profile of any of the infused groups, which goes along with the hypothesis of increased perfusion of PMN in the mammary. Still, the milk culture reveals no difference in the bacterial population. Perhaps more histological information and diverse markers of the adaptive immune response are needed to understand the integrity of a mammary gland and factors triggering the chronic elevated SCC. 

The isolated neutrophils’ phagocytic competence was also evaluated among groups. A previous investigation reported that bovine circulatory neutrophils treated with rbIL-8 presented greater phagocytic activity than untreated controls [16]. In the present study, the rbIL-8 group tended to present a greater number of neutrophils with high fluorescence for PI than the control, but it was similar to the saline group. This tendency is likely attributed to the fact that the chronic cows had a greater number of neutrophils compared to the control, based on flow cytometry (*p* = 0.052), rather than being an effect of the treatment itself. In fact, the contrast analysis for the number of neutrophils regarding high fluorescence for PI reveals that the chronic cows differed from the control (Figure 3A). 

Contrary to our hypothesis, the rbIL-8 group presented a similar phagocytic response compared to the saline group, marked by the log of neutrophils with high fluorescence mutually for PI and DHR (Figure 5A,B). In addition, the infused groups presented greater mean fluorescence for both dyes (Figure 6) and more high-fluorescence neutrophils in the milk compared with the control groups (Figure 5A), which could be partially attributed to the fact that the control group had fewer neutrophils isolated than the chronic groups (3.36 ± 0.18 vs. 3.87 ± 0.09, respectively). Another study reported that cows with mastitis have a greater number of neutrophils positive for ROS compared to non-mastitis cows after being challenged with *Staphylococcus aureus* [21]. Perhaps, neutrophils from chronic mastitis cows are in an active state and are capable of eliciting phagocytosis but do not resolve the infection. This non-resolution status might lead to more neutrophil recruitment and deleterious tissue damage to the alveoli due to the presence of necrotic PMN [22]. One of the features that could elucidate the prolonged inflammatory state in chronic mastitis is the evaluation of neutrophil apoptosis. Although phagocytosis was similar between infused groups, the number of neutrophils undergoing apoptotic processes could be distinct among groups, which could influence the overall immune competence of the affected quarter to an insult. In the present study, part of the flow-cytometry protocols included using dyes for caspase-3 and 6 (markers of apoptosis [23]) in both specimens. Preparation of the protocol using dyes specific to apoptosis failed, and the assessment of neutrophil apoptosis was discarded, configuring the inability of the current investigation to evaluate apoptosis in the isolated neutrophils.

Contrary to our hypothesis, the circulatory neutrophils’ phagocytosis ability was similar between the infused groups, showing that the intramammary treatment with rbIL-8 was not different from saline in terms of inducing neutrophil recruitment. Additionally, when comparing the chronic groups with the control, the circulatory concentrations of neutrophils and mean fluorescence were similar, indicating that the chronic mastitis and non-mastitis cows had a similar phagocytic and recruitment ability. A previous investigation that treated post-partum cows with an intrauterine infusion of rbIL-8 reported an increase of circulatory IL-8 of up to 350 pg/mL in three days relative to treatment, which was significant compared to untreated controls [24]. It is noteworthy that in the present study, the circulatory concentrations of IL-8 were not quantified, which limits the assumption that the intramammary treatment of rbIL-8 reached the bloodstream, especially in chronic mastitis cows in which the blood–epithelial interface of the mammary gland alveoli might be disrupted [22]. Overall, both the circulatory neutrophils and milk neutrophils revealed similar phagocytosis responses to the infused treatments, but only in an acute manner. It is unknown if the neutrophil phagocytosis response would change past 48 h, since a shift in PMN profile (band to segmented neutrophils) is also reported at 72 and 78 h post intramammary and intravenous injection, respectively, of *E. coli* endotoxin [25,26]. The study power analysis was performed considering differences in SCC concentration in the milk, not in the circulatory PMN profile. Hence, the present study might be underpowered to differentiate shifts in the PMN cells of the bloodstream, which may explain the lack of statistical difference in the circulatory number of neutrophils among the three groups.

Biochemical profile was also characterized for each group, and the total protein was similar between the infused groups, but the globulins tended to differ between them. In fact, rbIL-8-treated cows had higher circulatory concentrations of globulins than saline. Immunoglobulins such as IgM and IgG are the main opsonizing antibodies for PMN, including in cattle [27]. However, the IgG concentrations quantified from milk samples were similar among groups. In dry-off cows treated with 25 µg of intramammary rbIL-8, IgG concentrations were increased in the mammary secretions [17]. In the present study, maybe the rbIL-8 could increase the circulatory globulin concentration. Still, the globulin did not cross the disrupted blood-barrier alveoli, remaining elevated only in the bloodstream. Haptoglobin was also quantified in the milk samples and was similar among all groups. This acute phase protein is reported to increase more than 10 folds in milk from a mastitis cow compared to healthy ones [28]. One of the sources of haptoglobin in milk is from alveoli neutrophils and epithelial cells [29]. The similarity of this protein concentration in milk among groups might be attributed to different reasons. First, although haptoglobin concentrations increase in mastitis milk, it presents a wide range (from 2.08 to 55.46 µg/mL) when the SCC is above 200 × 10^3^ cells/mL, which may overlap with the concentrations of healthy and sub-clinical mastitis cows [30]. Second, most likely, the local neutrophils did not induce the acute response expected (especially for the rbIL-8) to increase the haptoglobin concentrations. Circulatory concentrations of CK were increased in the saline-treated cows compared with the rbIL-8 ones at 4 h related to treatment. This enzyme converts creatinine to creatine phosphate and has high activity in muscles and the brain [31]. The enzyme CK is also reported as increased in cows diagnosed with displaced abomasum [32]. Nevertheless, none of the cows enrolled in the study presented changes in locomotion, feeding, or water consumption, at any time during the experiment. Hence, the reason for this difference in the circulatory concentration of CK is unclear. 

## 4. Conclusions

In conclusion, the intramammary infusion of rbIL-8 did not resolve the chronic elevated SCC condition in dairy cows. The SCC concentrations in the milk samples collected from the rbIL-8 infused group at 0 h were not statistically significant compared with 144 h relative to treatment. As expected, the rbIL-8 treatment elicited an acute elevation of SCC post intramammary infusion of the molecule, but the persistent elevation of the SCC from 12 to 48 h relative to treatment noticed was not statistically different from the saline-treated group. Another aim of the study was to evaluate the neutrophil’s phagocytosis capability and dynamics in blood and milk specimens. However, contrary to our hypothesis, the total amount of neutrophils isolated, phagocytosis ability (dictated by the number of neutrophils with high or low fluorescence), and mean fluorescence of propidium iodide and dihydrorhodamine were unaltered in the rbIL-8 group compared with the saline intramammary infusion. Another result contrary to the initial hypothesis is the milk concentration of haptoglobin and IgG, which is similar between the two infused groups. The haptoglobin concentration relative to hour post-treatment was not statistically different from the control cows. The cows intramammary infused with rbIL-8 presented altered circulatory concentrations of total proteins, globulin, and albumin when compared with the control group. However, those circulatory proteins were similar to the saline group concentrations. Perhaps the condition of cows presenting chronic elevated SCC is irreversible due to potential tissue damage caused by persistent mastitis or might require repeated treatments to elicit a prolonged response capable to change the chronic issue. Another alternative for using rbIL-8 is on acute mastitis cases, but this requires investigation.

## 5. Materials and Methods

All experimental procedures involved in this present study were approved by the Institutional Animal Care and Use Committee of the University of Illinois Urbana-Champaign (Protocol # 19106).

### 5.1. Animal Selection, Housing, and Treatments

Holstein lactating multiparous cows (n = 8), with at least one of the quarters with a SCC > 300,000 cells/mL for at least 3 consecutive months (average SCC = 2957 ± 1698 × 10^3^ cells/mL), free of clinical mastitis signs and concurrent diseases, were purchased from a local farm and housed at the University of Illinois Dairy Farm. All the cows were placed in a free-stall barn two weeks before starting the treatment and were milked and fed twice a day with a total mix ratio formulated on the farm. Each quarter, from each cow, was individually evaluated for quantification of milk SCC, using a portable DeLaval cell counter, according to the manufacturer’s instructions (DCC; DeLaval International, Lund, Sweden). The quarter that contained the highest number of SCC was chosen as the treatment quarter for the respective cow. The chronic high SCC cows were randomized based on the SCC. A milk sample from the selected quarter was sent to the University of Illinois Diagnostics Laboratory for a mastitis culture pre-and post-treatment. The pre-treatment bacteriological profile revealed a variety of species of bacteria, including *Klebsiella* sp., *Staphylococcus* sp., *Streptococcus* sp., *Pseudomonas* sp., and *Corynebacterium* sp., for all treatment cows. The treatment groups were classified as rbIL-8 (n = 4, SCC = 3259 ± 1625 × 10^3^ cells/mL), receiving a 10 mL solution containing 1 mL of rbIL-8 (1.57 mg/mL), developed by our research group (Bicalho et al., 2019), with 9 mL of 0.9 % sodium chloride as the excipient. The other treatment group was classified as saline (n = 4, SCC = 2657 ± 1962 × 10^3^ cells/mL), receiving 10 mL of the same excipient as the other group. Both treatments were applied once (established as 0 h) for the selected quarter, slowly injected using a sterile syringe containing the respective solution, immediately after the morning milking while animals were still in the parlor. An untreated group (control, n = 2, with SCC < 300.000 cells/mL) was also evaluated concurrently with the treatment groups. 

### 5.2. Somatic Cell Count, PMN Isolation from Milk and Blood Samples

All milk samples were collected aseptically by using an alcohol wipe in the teat-end, discarding the first three strips before the sample was collected. Quantification of milk SCC was performed at 0, 4, 8, 12, 24, 48, and 144 h relative to treatment, via DeLaval cell counter using disposable cassettes for each sample collected (DeLaval International,Lund, Sweden). Briefly, milk samples were collected in a 15 mL falcon tube, homogenized, stored in ice (in a rack), and transported to the lab, where the SCC was quantified and recorded. Blood samples were collected via venipuncture of the coccygeal vessels, using a 10 mL heparin tube for biochemical profile for each cow at 0, 4, and 24 h relative to treatment, and 10 mL in EDTA tubes at 0, 4, 8, 24, 48 h relative to treatment, for white blood cell count. The veterinary diagnostic laboratory of the University of Illinois conducted the biochemical profile using the Clinical Chemistry Analyzer AU680 (Beckman Coulter Inc, Brea, CA, US), and a white blood cell count on a cell counter. Commercial ELISA kits were used to assess milk haptoglobin (Life Diagnostics, Inc., West Chester, PA, USA. Catalog Number: HAPT-11) and IgG (Bethyl Laboratories, Montgomery, TX, USA. Cat. No. E11-E118) concentrations at 0, 4, 8, 24, 48 h relative to treatment.

Neutrophils were isolated from blood and milk samples to assess phagocytic activity and respiratory oxidative burst via flow cytometry at 0, 24, and 48 h post-treatment. The protocol was similar to that described in previous work [33] and was concluded in less than two hours post-sampling. Briefly, 50 mL of a milk sample was collected immediately before the morning milking, placed on ice, and transported to the laboratory. The sample was filtered using a 40 µm cell strainer (Sigma-Aldrich, San Louis, MO, USA/CLS431750) to remove the milk fat, and then it was diluted with cold (4 °C) cell-culture-graded calcium- and magnesium-free PBS (Fisher-Scientific, Waltham, MA, USA/MT21040CV). The diluted sample was then centrifuged at 600× *g* for 15 min at 4 °C, and the remaining fat was removed carefully using a 5 mL transfer pipette. A pellet was formed in the bottom of the tube, and it was washed twice at 300× *g* for 10 min and 200× *g* for 15 min at 4 °C. The remaining cells were resuspended with DPBS (Fischer Scientific/MT21031CV) supplemented with gelatin 0.5 mg/mL (Sigma-Aldrich/G9382) at room temperature. Two different density Percoll gradients (Sigma-Aldrich/P4937) were prepared (1.092 and 1.071) and layered in a 10 mL test tube, to isolate the cells by gradient difference. Then, 3 mL of the milk suspension was overlaid on the Percoll gradients and centrifuged at 400× *g* for 25 min at 4 °C. Isolated cells were mostly in between the gradients, but in some samples, they were also in the lower portion of the tube. Homologous plasma was added to the isolated cells solution in a 1:1 ratio to prevent lysis during the preparation of smears. Similarly, blood samples (10 mL) were collected in EDTA tubes and placed in a rack inside a cool box filled with ice. It is noteworthy that the blood samples were not in direct contact with the ice, to avoid PMN degeneration. Plasma and buffy coat portions of the blood samples were extracted after centrifugation and supplemented with gelatin at 0.5 mg/mL. The cell isolation was conducted as mentioned above using the two Percoll gradients, and the same isolates were collected. The assumed PMN was resuspended in RPMI 1640 until the flow cytometry reagents were added. Slide smears were prepared using the isolated cells of both specimens to differentiate the PMN population (primarily composed of neutrophils), and cell count was performed using a hemocytometer chamber with 10 uL cells suspension in a 1:1 ratio with 0.4 % trypan blue, for quantification of PMN necessary for loading the flow cytometry dyes. 

### 5.3. Escherichia Coli Labeled Propidium Iodide and Flow Cytometry

A milk sample containing a positive culture for Escherichia coli was used as a bacterial source to challenge the neutrophil’s immune response in vitro. The culture, isolation, and bacteria labeling were performed similarly to those described elsewhere [34,35]. Briefly, cultures were grown in tryptic soy broth and then diluted with 20% glycerol. The bacteria were inactivated under heat at 56 °C for 30 min, homogenized, and split into 10 mL vials for labeling. Each tube was centrifuged at 2200 rpm for 30 min. After centrifugation, the supernatant was discarded, and the bacteria were resuspended with 9 mL of PBS + 1 mL of propidium iodide (PI). The vial was wrapped in aluminum foil to protect it from light and kept on continuous rotation overnight at room temperature for labeling. Twelve hours later, the vials were centrifuged at 960× *g* for 30 min at 23 °C and resuspended in 10 mL of PBS twice to extinguish the excess propidium iodide in the media. The propidium-iodide-labeled bacteria were protected from light and stored at 5 °C until use, and the final concentration of the *E. coli*-labeled PI solution was 5 × 10^8^ cells/mL. 

Samples were prepared as follows. First, 100 µL of isolated PMN, of both specimens, was loaded with 10 μL of a 50 μM dihydrorhodamine 123 solution for 10 min at 37 °C with constant rotation. Then, the PI-labeled *E. coli* were added to the samples in a 40:1 bacteria-to-neutrophil ratio. Tubes were vortexed for about a second and incubated at 37 °C for 30 min with continuous mixing. After incubation, tubes were placed on ice in order to cease the neutrophils’ phagocytosis and oxidative burst. Controls were used to guide the flow cytometry for the positive dye signals: a tube containing only cells (100 µL), a negative control tube containing dihydrorhodamine 123 (DHR)-loaded leukocytes without PI-labeled bacteria (100 µL + 0.5 µM of DHR), a positive control for DHR containing phorbol 12-myristate 13-acetate (PMA, induces oxidative burst in neutrophils) without PI-labeled bacteria (100 µL + 0.5 µM of DHR +10 µL of PMA), and for PI, dead cells loaded with PI (100 µL + 0.5 µL of PI ). The flow cytometry machine used was a Cytek Aurora (Cytek Biosciences) containing a red, blue, and violet laser, and the respective excitation sources were 80 mW at 640 nm, 50 mW at 488 nm, and 100 mW at 405 nm. Density cytograms were created using the linear amplification of the signals in the forward and side scatter channels. The fluorescence cytograms were based on the amplification of the fluorescence signal of the cells as it crosses throughout the laser beam. Neutrophils were selectively analyzed based on their size and complexity in the density cytogram, and acquisition data was measured based on a stopping time of 5 min (15 µL/min) per sample, processed by the computer software SpectroFlo. The total amount of cells, high- and low-fluorescent neutrophils stained for PI (red) or DHR (blue and violet), and those for both dyes were recorded in a mixed panel, based on the fluorescence for each quadrant. The additional analysis included the mean fluorescence intensity for PI-tagged cells, as well as the mean fluorescence intensity for DHR-tagged cells. 

### 5.4. Statistical Analysis 

Power analysis was conducted in SAS 9.4 (SAS/ STAT, SAS Inst. Inc., Cary, NC, USA), based on the log SCC mean difference expected between groups (4 units) and standard deviation (2 units), using α = 0.05 and β = 0.20, in one-tailed *t*-test. A total of 8 cows were deemed necessary with a power of 0.802. Expected mean differences were estimated based on a previous investigation that reported an increase in the log SCC of about 3 units in the first 24 h after treating dry-off cows with 25 ug/mL of rb-IL8 intramammary (Watanabe et al., 2008). Since the cows in the present study will be lactating and the treatment dose is 1.57 mg/mL, 4 units of change in the log SCC is a conservative estimate effect of the treatment. 

The statistical analysis was conducted in SAS 9.4, using a mixed model procedure. The models included the treatment, hour, and the interaction of treatment by hour. Spatial power is the variance-covariance structure of choice given the nature of the data (unequally spaced hours relative to treatment and heterogeneous variance among distinct time points) and the smallest Shwarz’s Bayesian information. Repeated measures were considered for hours relative to treatment, and the animal identification was nested within the treatment and assigned as a random effect. The Tukey–Kramer method was used for the adjustment of multiple comparisons for all the responses when the three groups were compared. Orthogonal contrast was applied to test the effect of SCC level, which consisted of the chronic groups versus the control (rbIL-8 + saline vs. control), and also to compare the effect between both treated groups, called infused comparison (rbIL8 vs. saline). If the contrast evaluation presented a statistical significance or tendency, the categories of the respective comparison were evaluated separately in another model, using the same model structures and procedures mentioned previously. Upon analysis of the separate model, if the treatment of the interaction of treatment and hour showed a statistical significance or tendency, the slice method was performed by the hour to evaluate the effect of the treatment separately for each hour. All response variables were checked for convergence criteria; if not, a compound symmetry structure was used. The normality of the residuals was checked for all response variables according to the Pearson and studentized residuals. Residuals following a non-normal distribution were transformed according to the Box Cox curve using macro. Response means were back-transformed to the original scale (except for the ones presented in the log) after the statistical analysis. Statistical significance was considered as *p* ≤ 0.05, and statistical significance as 0.05 < *p* ≤ 0.10. 

## Figures and Tables

**Figure 1 antibiotics-11-01029-f001:**
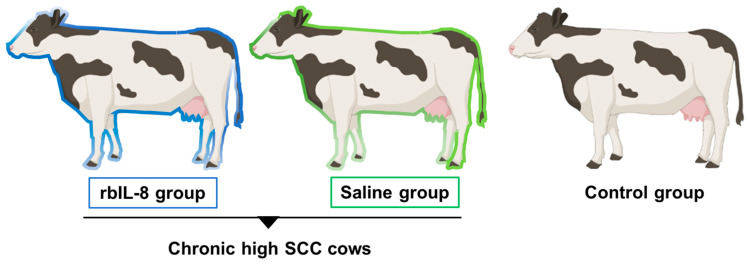
Visual representation of the cows presenting chronic high SCC and the low SCC cows (control). The chronic cows were submitted to one of two intramammary treatments, rbIL-8 (blue outline) or saline (green outline), whereas the control group remained untreated (no outlines). The chronic high SCC cows were randomized based on the quarter that presented the highest SCC, and only the respective quarter received the intramammary infusion. Both rbIL-8 and saline groups had a similar SCC at 0 h relative to treatment (rbIL-8: SCC = 3259 ± 1625 × 10^3^ cells/mL; and saline: SCC = 2657 ± 1625 × 10^3^ cells/mL, *p* = 0.519).

**Figure 2 antibiotics-11-01029-f002:**
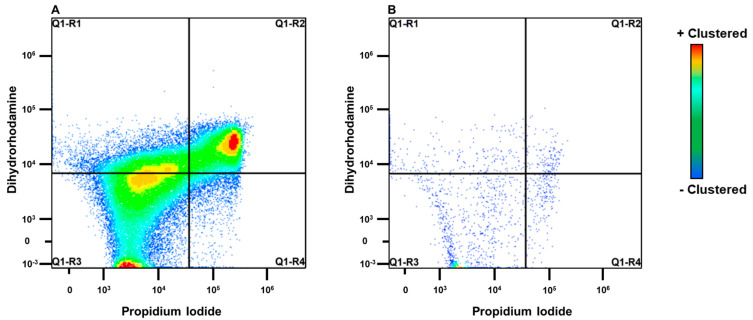
Representative density plot on flow cytometric analysis for a chronic (**A**) and a control cow (**B**). The image is plotted specifically for neutrophils, after the isolation and the staining protocol. The quadrant representing neutrophils that presented low fluorescence for both propidium iodide (PI) and dihydrorhodamine (DHR) is indicated in Q1-R3. Neutrophils presenting high fluorescence only for PI are indicated on quadrant Q1-R4, and neutrophils presenting high fluorescence only for DHR are indicated on quadrant Q1-R1. The quadrant Q1-R2 indicates neutrophils presenting high fluorescence for both PI and DHR. On the Y and X axis, there is the fluorescence intensity for each dye.

**Figure 3 antibiotics-11-01029-f003:**
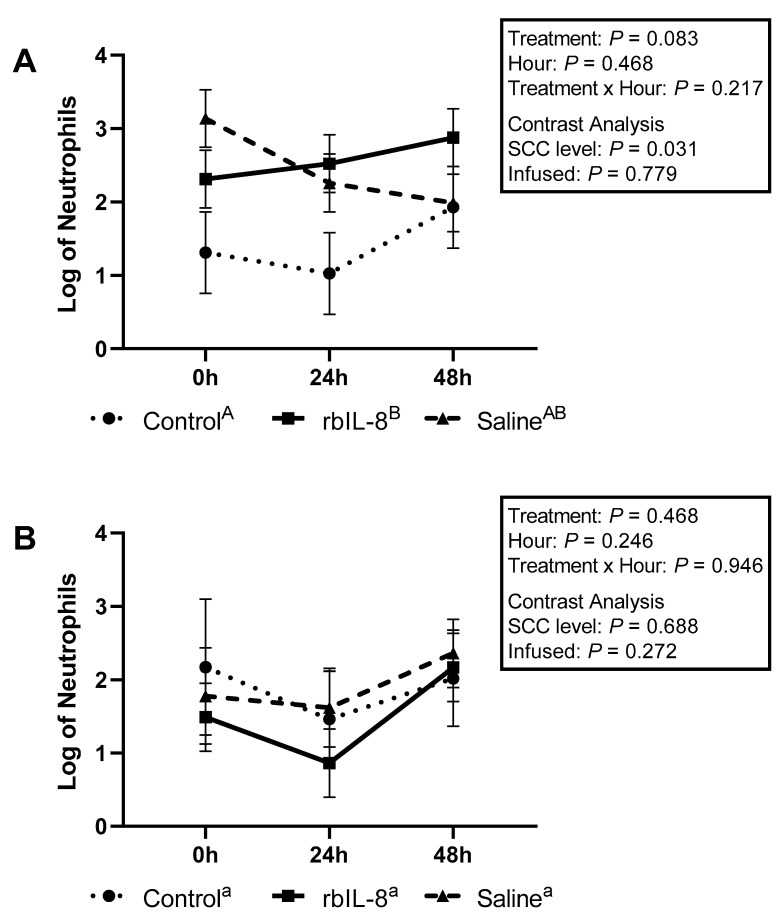
Log of neutrophils isolated from milk (**A**) and from blood samples (**B**) that presented a high fluorescence only for propidium iodide (PI) relative to hours post-treatment based on flow cytometry. The treatment effect refers to multiple comparisons among the groups. The SCC level was performed using a contrast comparing the chronic groups (rbIL-8 and saline) versus control, whereas the infused was compared only the rbIL-8 with the saline. Multiple comparisons were adjusted for Tukey–Kramer. Groups with different superscripts ^a,b^ differed, and ^A,B^ presented a tendency for the mean ± SE of log of neutrophils, in the multiple comparison analysis; *p* = ≤0.05 means a statistical significance, and 0.05 < *p* ≤ 0.10 means a statistical tendency.

**Figure 4 antibiotics-11-01029-f004:**
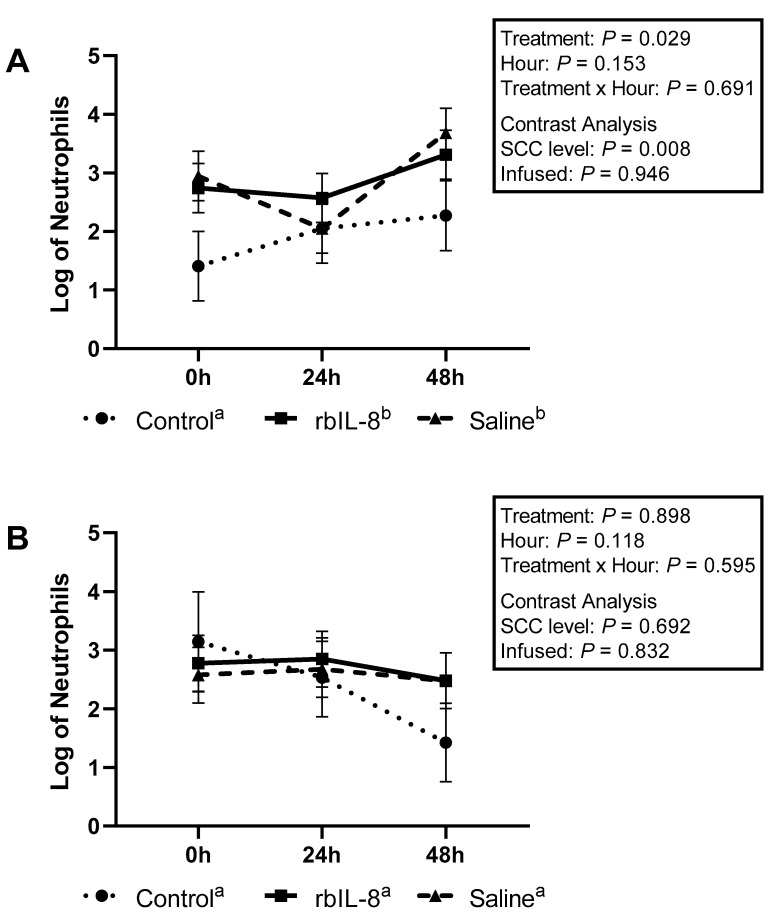
Log of neutrophils isolated from milk (**A**) and from blood samples (**B**) that presented a high fluorescence only for dihydrorhodamine 123 (DHR) relative to hours post-treatment based on flow cytometry. The treatment effect is a multiple comparison analysis among the groups. The SCC level was measured using a contrast comparing the chronic groups (rbIL-8 and saline) to the control, whereas the infused was compared to the rbIL-8 with the saline. Multiple comparisons were adjusted for Tukey–Kramer. Groups with different superscripts ^a,b^ differed for the mean ± SE of log of neutrophils, in the multiple comparison analysis; *p* = ≤0.05 means a statistical significance, and 0.05 < *p* ≤ 0.10 means a statistical tendency.

**Figure 5 antibiotics-11-01029-f005:**
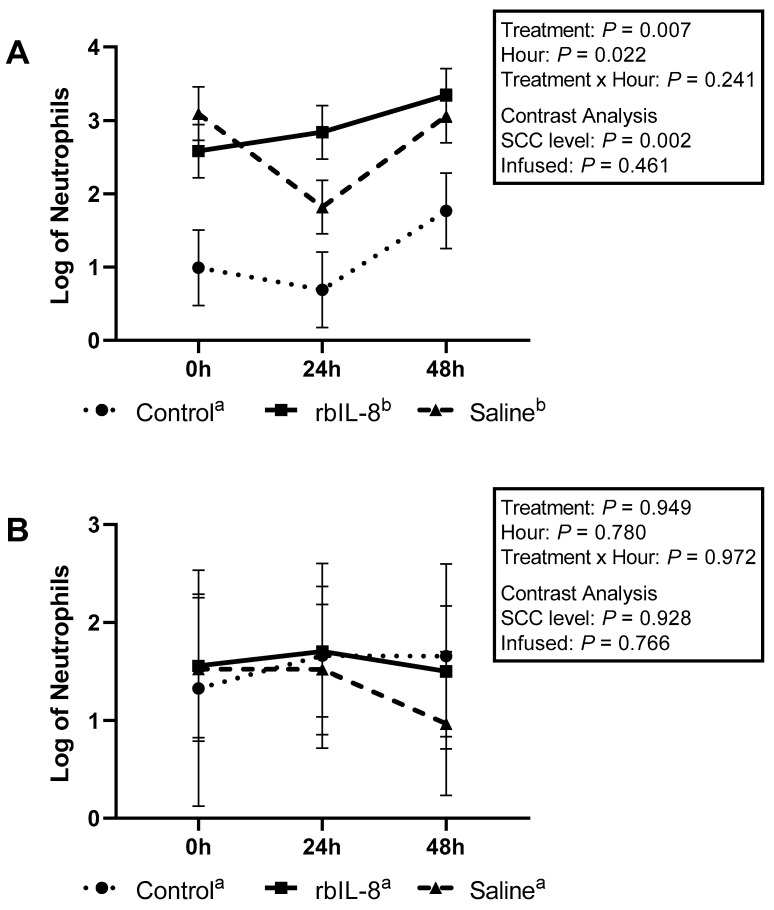
Log of neutrophils isolated from milk (**A**) and from blood samples (**B**) that presented a high fluorescence for both propidium iodide (PI) and dihydrorhodamine 123 (DHR) relative to hours post-treatment based on flow cytometry. The treatment effect is a multiple comparison analysis among the groups. The SCC level was measured using a contrast comparing the chronic groups (rbIL-8 and saline) to the control, whereas the infused was compared to the rbIL-8 with the saline. Multiple comparisons were adjusted for Tukey–Kramer. Groups with different superscripts ^a,b^ differed for the mean ± SE of log of neutrophils, in the multiple comparison analysis; *p* = ≤0.05 means a statistical significance, and 0.05 < *p* ≤ 0.10 means a statistical tendency.

**Figure 6 antibiotics-11-01029-f006:**
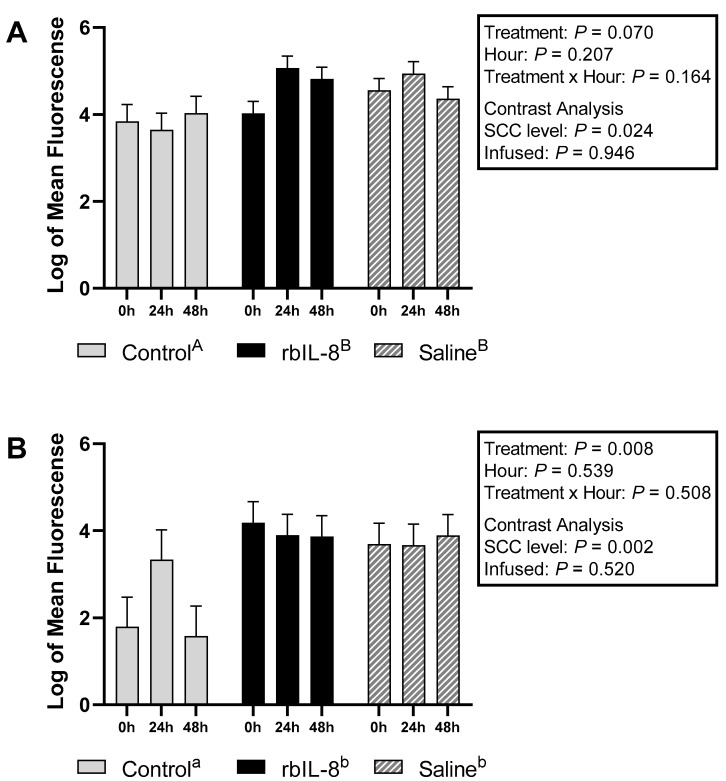
Mean fluorescence for neutrophils isolated from milk samples, for each respective dye, propidium iodide (**A**) and dihydrorhodamine (**B**), relative to hours post-treatment based on flow cytometry. Treatment effect is a multiple comparison among the groups. The SCC level was measured using a contrast comparing the chronic groups (rbIL-8 and saline) to the control, whereas the infused compares the rbIL-8 with the saline. Multiple comparisons were adjusted for Tukey–Kramer. Groups with different superscript ^a,b^ differed and ^A,B^ presented a tendency for the mean ± SE of log of neutrophils, in the multiple comparison analysis; *p* = ≤0.05 means a statistical significance, and 0.05 < *p* ≤ 0.10 means a statistical tendency.

**Figure 7 antibiotics-11-01029-f007:**
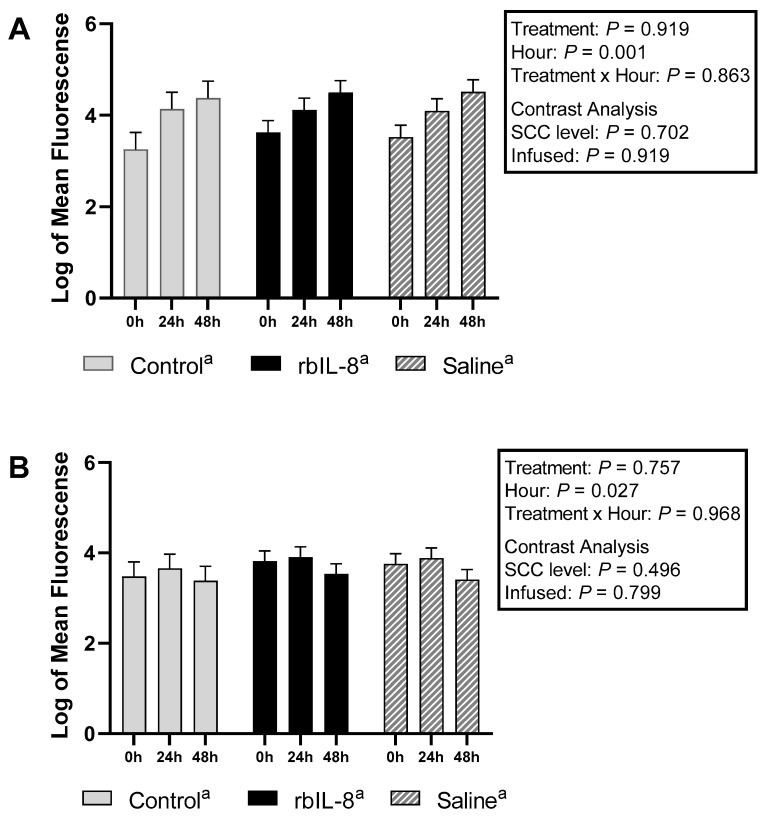
Mean fluorescence for neutrophils isolated from blood samples, for each respective dye, propidium iodide (**A**) and dihydrorhodamine (**B**), relative to hours post-treatment on flow cytometry. The treatment effect is a multiple comparison analysis among the groups. The SCC level was measured using a contrast comparing the chronic groups (rbIL-8 and saline) to the control, whereas the infused compared the rbIL-8 with the saline. Multiple comparisons were adjusted for Tukey–Kramer. Groups did not differ (superscripts ^a^) for the mean ± SE of log of neutrophils, in the multiple comparison analysis; *p* = ≤0.05 means a statistical significance, and 0.05 < *p* ≤ 0.10 means a statistical tendency.

**Figure 8 antibiotics-11-01029-f008:**
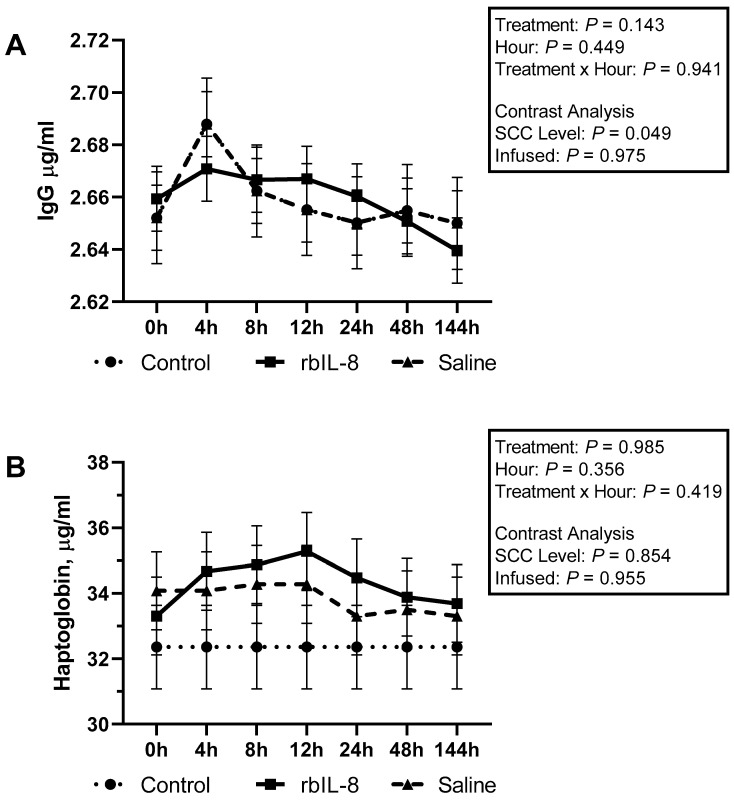
Concentrations of IgG (**A**) and haptoglobin (**B**) in milk relative to treatment for each group. Treatment effect is a multiple comparison among the groups. The SCC level was measured using a contrast comparing the chronic groups (rbIL-8 and saline) to the control, whereas the infused compares the rbIL-8 with the saline. Multiple comparisons were adjusted for Tukey–Kramer; *p* = ≤ 0.05 means a statistical significance, and 0.05 < *p* ≤ 0.10 means a statistical tendency.

**Figure 9 antibiotics-11-01029-f009:**
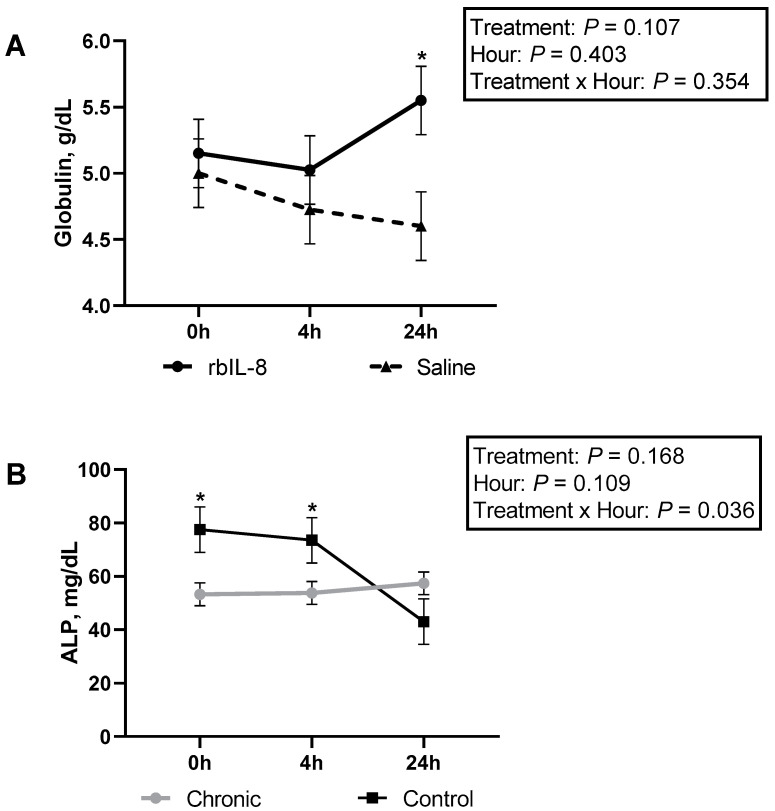
Circulatory concentrations of globulin (**A**) and ALP (**B**). In (**A**), the comparison between the infused groups (rbIL-8 vs. saline) is shown, and in (**B**), the comparison between the chronic (rbIL-8 and saline) vs. control, in hours relative to treatment. Significant time point differences (*p* < 0.05) between groups are represented with an asterisk (*).

**Figure 10 antibiotics-11-01029-f010:**
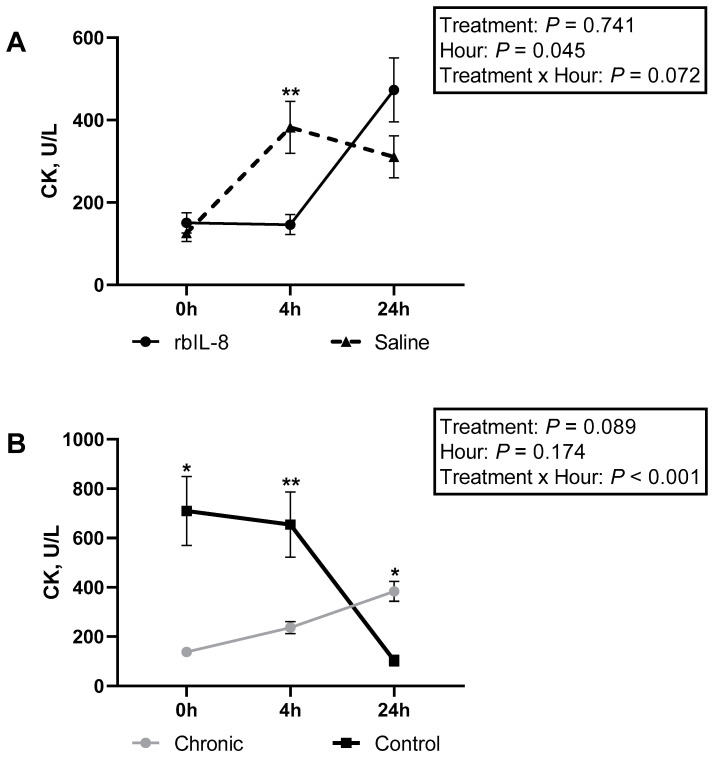
Circulatory concentrations of CK in (**A**,**B**). (**A**) shows the comparison between the infused groups (rbIL-8 vs. saline), and (**B**), the comparison between the chronic (rbIL-8 and saline) vs. control, in hours relative to treatment. Significant time point differences (*p* < 0.05) between groups are represented with an asterisk (*), and tendencies (0.05 < *p* ≤ 0.10) with double asterisk (**).

**Figure 11 antibiotics-11-01029-f011:**
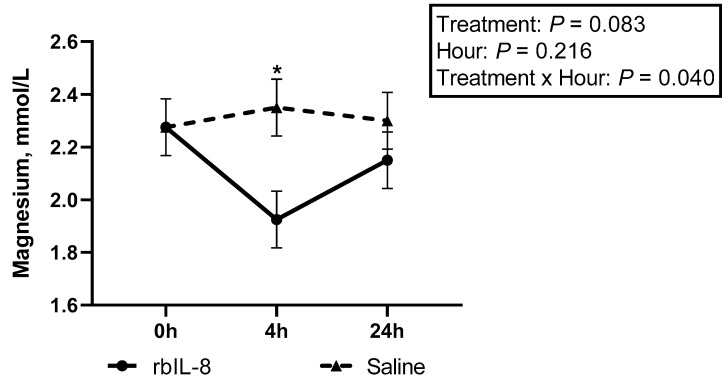
Circulatory concentrations of magnesium between the infused groups in hours relative to treatment. Significant time point differences (*p* < 0.05) between groups are represented with an asterisk (*).

**Table 1 antibiotics-11-01029-t001:** The WBC percentage change relative to treatment.

Item	Hours Relative to Treatment						Effect ^3^				
WBC ^1^ Δ%	0 h ^2^	0–4 h	4–8 h	8–24 h	24–48 h	Mean Δ%	Group	Hour	Group*Hour	Infused	SCC level
Control	9.6 ± 5.9	14.2 ± 13.6	−1.2 ± 13.6	−3.0 ± 13.6	6.1 ± 13.6	4.0 ± 6.1 ^a^	*p*-value				
rbIL-8	7.0 ± 4.2	−15.7 ± 9.4	19.2 ± 9.4	7.0 ± 9.4	17.3 ± 9.4	6.9 ± 4.3 ^a^	0.816	0.930	0.167	0.880	0.539
Saline	15. ± 4.2	15.5 ± 9.4	−1.2 ± 9.4	−0.04 ± 9.4	−1.6 ± 9.4	3.1 ± 4.3 ^a^					

^1^ White blood cell percentage change (Δ%) relative to an hour. ^2^ At 0 h, the WBC concentrations (mean ± SE) in 10^3^ cells/µL. ^3^ Multiple comparisons among groups, hour, and the interaction of group by hour. Contrast analysis of infused cows (rbIL-8 vs. saline) and the SCC Level (rbIL-8 and saline vs. control). Multiple comparisons were adjusted for Tukey–Kramer. Different superscripts ^a,b^ indicate that means ± SE differed among groups in the multiple comparison analysis; *p* = ≤0.05 means a statistical significance, while 0.05 < *p* ≤ 0.10 means a statistical tendency.

**Table 2 antibiotics-11-01029-t002:** The log of the somatic cell count (SCC) per mL concentrations relative to treatment.

Item	Hours Relative to Treatment										Effect ^2^		
LogSCC ^1^	0 h	4 h	8 h	12 h	24 h	48 h	144 h	Mean	Group	Hour	Group*Hour	Infused	SCC Level
Control	3.92 ± 0.2 ^a^	5.31 ± 0.2 ^a^	4.79 ± 0.2 ^a^	5.52 ± 0.2 ^a^	4.03 ± 0.2 ^a^	4.55 ± 0.2 ^a^	4.79 ± 0.2 ^a^	4.71 ± 0.1 ^a^	*p*-value				
rbIL-8	6.47 ± 0.1 ^b^	6.55 ± 0.1 ^b^	6.34 ± 0.1 ^b^	6.50 ± 0.1 ^a^	6.54 ± 0.1 ^b^	6.56 ± 0.1 ^b^	6.15 ± 0.1 ^b^	6.44 ± 0.1 ^b^	0.0002	<0.0001	<0.0001	0.369	<0.0001
Saline	6.31 ± 0.1 ^b^	6.49 ± 0.1 ^b^	6.36 ± 0.1 ^b^	6.45 ± 0.1 ^a^	6.07± 0.1 ^b^	6.18 ± 0.1 ^b^	6.17 ± 0.1 ^b^	6.29 ± 0.1 ^b^					

^1^ LogSCC in cells/mL, relative to treatment for each respective group (mean ± SE). ^2^ Multiple comparisons among groups, hour, and the interaction group by hour. Contrast analysis of the infused cows (rbIL-8 vs. saline) and the SCC Level (rbIL-8 and saline vs. control). Multiple comparisons were adjusted for Tukey–Kramer. Different superscript ^a,b^ indicates that means ± SE differed among groups in the multiple comparison analysis; *p* = ≤0.05 means a statistical significance, and 0.05 < *p* ≤ 0.10 means a statistical tendency.

**Table 3 antibiotics-11-01029-t003:** Biochemical profile among groups and hours relative to treatment.

	Hours Relative to Treatment						Effect ^1^		
Item	0 h	4 h	24 h	Mean	Group	Hour	Group*Hour	Infused	SCC Level
Total protein, g/dL					*p*-value				
Control	7.8 ± 0.25	7.8 ± 0.25	8.2 ± 0.25	7.9 ± 0.17 ^a^					
rbIL-8	8.5 ± 0.17	8.3 ± 0.17	8.8 ± 0.17	8.5 ± 0.12 ^b^	0.054	0.016	0.430	0.161	0.037
Saline	8.4 ± 0.17	8.2 ± 0.17	8.2 ± 0.17	8.3 ± 0.12 ^ab^					
Albumin, g/dL									
Control	3.9 ± 0.17	3.8 ± 0.17	3.4 ± 0.17	3.7 ± 0.11 ^a^					
rbIL-8	3.4 ± 4.2	3.3 ± 0.12	3.2 ± 0.12	3.3 ± 0.07 ^b^	0.062	0.433	0.278	0.130	0.056
Saline	3.4 ± 0.12	3.5 ± 0.12	3.6 ± 0.12	3.5 ± 0.07 ^ab^					
Globulin, g/dL									
Control	15.3 ± 3.4	16.1 ± 3.4	23.6 ± 3.4	18.3 ± 2.3 ^a^					
rbIL-8	26.6 ± 2.4	25.3 ± 2.4	30.8 ± 2.4	27.6 ± 1.6 ^b^	0.032	0.202	0.275	0.106	0.025
Saline	25.9 ± 2.4	22.5 ± 2.4	21.6 ± 2.4	23.3 ± 1.6 ^ab^					
GGT, U/L									
Control	18.1 ± 4.0	20.0 ± 4.9	16.6 ± 3.4	18.1 ± 2.8 ^a^					
rbIL-8	26.6 ± 6.1	26.6 ± 6.1	33.3 ± 9.6	28.5 ± 5.0 ^a^	0.196	0.946	0.856	0.926	0.079
Saline	30.7 ± 8.2	28.5 ± 7.1	25 ± 5.4	27.9 ± 4.8 ^a^					
CK, U/L									
Control	709 ± 183	654 ± 183	103 ± 183	488 ± 129 ^a^					
rbIL-8	156 ± 129	153 ± 129	691 ± 129	333 ± 91 ^a^	0.551	0.518	0.013	0.294	0.897
Saline	131 ± 129	428 ± 129	389 ± 129	316 ± 91 ^a^					
Magnesium, mmol/L									
Control	2.7 ± 0.15	2.6 ± 0.15	2.2 ± 0.15	2.5 ± 0.10 ^a^					
rbIL-8	2.2 ± 0.10	1.9 ± 0.10	2.1 ± 0.10	2.1 ± 0.07 ^b^	0.028	0.096	0.036	0.085	0.027
Saline	2.2 ± 0.10	2.3 ± 0.10	2.3 ± 0.10	2.3 ± 0.07 ^ab^					

^1^ Multiple comparison analysis among groups, hour, and the interaction group by hour. Contrast analysis compared the infused cows (rbIL-8 vs. saline) and the SCC level (rbIL8 and saline vs. control). Multiple comparisons were adjusted for Tukey–Kramer. Different superscript ^a,b^ indicates that mean ± SE differed in the multiple comparison analysis; *p* = ≤0.05 means a statistical significance, and 0.05 < *p* ≤ 0.10 means a statistical tendency. Clinical Chemistry Analyzer AU680 (Beckman Coulter Inc, Brea, CA) was used to determine each metabolite concentration.

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
