# Peer review of "Evaluation of Recombinant Bovine Interleukin-8 (rbIL-8) as a Treatment for Chronic Intramammary Infection in Dairy Cows"

_antibiotics, 2022, doi:10.3390/antibiotics11081029_

Round 1

Reviewer 1 Report

Dear Authors,

Please check the attached file for my comments and concerns.

Kind regards

Author Response

AU: We appreciate very much the effort and insightful suggestions of the reviewers to improve the quality of the manuscript. We addressed the suggestions in the revised version of the manuscript. Please see below the answers to each comment after the abbreviation for authors (AU).

Reviewer 1

One background sentence before the objectives are recommended (L14)
AU:
Thanks for your suggestion. Changes were made as suggested.

The First mention of PI and DHR should be full (L2), PMN (L57), and in every instance
AU:
Thanks for catching that oversight. Changes were made as suggested.

L18-19: control cow SCC should also be >300000/ml. Changing the control group might change the result and subsequent discussion of the manuscript

AU: The untreated group with a SCC of less than 300,000 cells/ml was deliberately chosen to represent the results of an untreated group of animals with a healthy udder and low somatic cell count. The untreated group is a proxy of comparison between healthy and chronic cows with an “appropriate response” for SCC, neutrophils phagocytosis, etc. Both chronic groups (rbIL-8 and Saline) did not present an SCC concentration similar to that observed in healthy cows presenting low SCC, and part of the goal of the study was to represent the group and how it would impact the responses measured.

L55-63: background of rbIL8 should write more elaborately especially the effects (positive and negative) of IL8 on different body systems

AU: Thanks for your suggestion. Changes were made as suggested (L57-L77).

I am doubting the authors did not run the statistical analysis properly (L467-468). Typically, for a pairwise comparison with one statistical test, there should be only one P value. And after post-hoc analysis, the significant difference should be labeled with different letters for the compared means.

AU:  In the present study, there are three different groups 1) chronic cows for high SCC treated with rbIL-8, 2) chronic cows for high SCC treated with saline (negative control to assess the response of treatment), and an untreated Control group (low SCC that serve as a control for lack of SCC). Therefore, as a result of the experimental design used, a multiple comparison analysis is needed to assess significant mean differences among the three groups: rbIL-8 vs. Saline, rbIL-8 vs. Control, and Saline vs. Control. Whenever there is a difference in treatment in my analysis, it means that at least one of the groups differed from the other(s). The multivariable analysis will generate a P-value for each comparison, which is adjusted for Tukey-Kramer (a control method for type I error in multiple comparison analysis). We went ahead and added P-values for each of the multiple comparisons in the figure legends and the text to make it clearer to readers. Moreover, the contrast analysis answers different questions: 1) Effects of SCC level: Chronic cows (mean of rbIL8 + mean of Saline) vs. Control; and 2) Effects of treatment: rbIL8 vs. Saline.  When contrast is performed for both groups that were treated (rbIL-8 and saline), it removes the variability generated by the Control group in the multiple comparison analysis, which makes the test more sensitive to detect differences solely between rbIL8 vs. Saline. The comparison of rbIL8 vs. Saline in the multiple comparison analysis is different from the rbIL8 vs. Saline contrast, resulting in a different P-value.

For example, in this figure, Treatment is significant, according to multiple comparisons (the mean of at least one group differed from another). But upon the contrast analysis, it is revealed that this significant effect treatment in the log of neutrophils, coming from the chronic groups versus the control (P = 0.002), and not from the comparison rbIL8 vs. Saline (P = 0.4612). Contrast evaluation provides more information on how the chronic cows respond to the experiment compared to non-mastitis cows. Additionally, it also performs a unique comparison between the two infused groups (rbIL-8 vs. Saline).

Presentation of results is very complex.

AU: Thanks for your comment. We agree that there are several results and the task of sharing it concisely and clearly is challenging. Reviewers’ concerns about the complexity of the results were addressed. For example, an explanation for the contrast in all the legends of the figures in which it is present and on the materials and methods section. Results of WBC and SCC are now presented in separate tables (Tables 1 and 2). The mean fluorescence results are in 4 figures instead of two. We expect that dividing the data into smaller chunks of information with more detailed statistical methods in each figure will improve the readability of the results section.

Discussion is not sufficient 
The discussion was expanded to cover more broadly the responses explored in the study.

Conclusion is very brief

From our perspective, the conclusion must be limited to highlighting the main findings of the current study. A few other details were added. We are not sure about what the reviewer recommended when mentioning: “the use of statistical results in conclusion”. We briefly pointed out what is significant in terms of statistical results for most of the findings. Still, we have not included P-values and other statistical feature that is usually not an aspect expected to be included in a manuscript. As examples, we added below two other manuscripts recently published by the journal, that did not post any statistical results in conclusion:
Antibiotics 2021, 10, 490. https://doi.org/10.3390/antibiotics10050490

Antibiotics 2022, 11(7), 963; https://doi.org/10.3390/antibiotics11070963
We are assertive that the conclusion section in the present manuscript is clear and represents with precision the key findings identified by the study.

Reviewer 2 Report

The materials and methods lack methods for evaluating the biochemical profile of Albumin, Globulin , GGT, CK cholesterol, etc

The legends for figures 5, 6 and 7 need to be reformulated, the legend need to be self-explanatory.

Figure 4 must be reformulated is confusing the legend too.

Line 352 would be good to spell out the abbreviation “TMR”

In topic 5,1 “Animal selection, housing and treatments”. the authors do not indicate whether the animals are known to have a history of subclinical mastitis; if the cows are multiparous this information should be part of the exclusion and inclusion criteria because the animals can change the immune response with each milk production campaign. Authors could include this information, because this information is very important.

Also In the experimental design the authors could have compared quarters of the same animal.

Author Response

Reviewer 2

The materials and methods lack methods for evaluating the biochemical profile of Albumin, Globulin, GGT, CK cholesterol, etc
AU: Thanks for your inquiry. The information is available on line 403 of materials and methods. Items in table 3 were evaluated using a standard automated Biochem Analyzer used by the Veterinary Diagnostic Laboratory of the University of Illinois.

The legends for figures 5, 6 and 7 need to be reformulated, the legend need to be self-explanatory.
AU:
Changes were made as suggested

Figure 4 must be reformulated is confusing the legend too.
AU:
Changes were made as suggested. The two figures were split into four, making the results more intuitive.

Line 352 would be good to spell out the abbreviation “TMR”
AU: Changes were made as suggested.

In topic 5,1 “Animal selection, housing and treatments”. the authors do not indicate whether the animals are known to have a history of subclinical mastitis; if the cows are multiparous this information should be part of the exclusion and inclusion criteria because the animals can change the immune response with each milk production campaign. Authors could include this information, because this information is very important.

AU:
The information requested is mentioned in the section “3.1 Animal selection, housing and treatments” of materials and methods. It reads “Line 373: Holstein lactating multiparous cows (n = 8), with at least one of the quarters with an SCC > 300,000 cells/ml for at least 3 consecutive months (average SCC = 2957 ± 1698 x103 cells/ml).”

 All the treated animals were multiparous and were all above the SCC threshold for at least three consecutive months. Therefore, the animal enrolled in the study and treated with intramammary infusion of rbIL-8 or saline followed the exact same criteria.

Also In the experimental design the authors could have compared quarters of the same animal.
AU:
Thanks for your question. It is a valid point. Perhaps future studies could use this approach to see if the intra-animal difference in SCC concentrations can be restricted to a quarter. In the current study not all cows have more than one quarter that would fit the criterion of high SCC and did not allow us to perform intra-animal comparison adequately. However, researchers must the cognizant that simple manipulation of the quarter to collect samples may lead to fluctuations in the SCC concentrations (as we observed in the untreated control in the present study), which may lead to confounded outcomes. A question that can be derived from this includes is the increase in SCC (or other features) on the control quarters caused by manipulation or an indirect effect of the infused quarter? To avoid any misleading information, we pursued the design with the current approach.

Round 2

Reviewer 2 Report

In the experimental design, the authors stated that they are multiparous cows, but did not include crucial information to assess the quality of the work, as this information should be taken into account as an exclusion/inclusion criterion for the animals; another issue is why the authors did not compare quarters of the same animal since each mammary quarter behaves as if it were an independent individual.

materials and methods must be improved for example  methods for evaluating the biochemical profile of Albumin, Globulin, GGT, CK cholesterol, must be better described including kits and equipment in order to increase the reproducibility of the work, flow citometry methods  could be better described including results section 2,1

in section 2,2 authors could made figures for the results of the first and second paragraph

the authors could include the mastitis history of the animals used in the present work, even to justify the non-comparison between mammary quarters of the same animal.

the authors could describe how the number of animals to be tested was decided, even because the comparison of parameters is between animals and not intra-animal.

Author Response

Title: Evaluation of recombinant bovine interleukin-8 (rbIL-8) as an antibiotic alternative for the treatment of chronic intra-mammary infection in dairy cows

Manuscript ID: antibiotics-1810298

Reviewer 2

In the experimental design, the authors stated that they are multiparous cows, but did not include crucial information to assess the quality of the work, as this information should be considered as an exclusion/inclusion criterion for the animals.

AU:
We would be happy to specifically address this concern, but remains unclear to the authors what “crucial information” the reviewer would like added to the manuscript. We were certain that these cows were truly presenting chronic high SCC during the last 3 months and that they didn’t possess any concurrent health disorders or conditions that could impact their immune system. After being purchased, these cows were randomized to each treatment group based on the animal’s SCC (information added to the materials and methods and in figure 2).

Another issue is why the authors did not compare quarters of the same animal since each mammary quarter behaves as if it were an independent individual.

AU: We addressed this matter in our previous review at length. Please see below our previous response. Albeit quarters might have a different number of SCC and presence of microbes that is not always the case and in the current study, many quarters would not fit the criteria (>300,000 cell/ml) to be used for the study.  

Previous response:

“AU: Thanks for your question. It is a valid point. Perhaps future studies could use this approach to see if the intra-animal difference in SCC concentrations can be restricted to a quarter. In the current study, not all cows have more than one quarter that would fit the criterion of high SCC and did not allow us to perform intra-animal comparison adequately. However, researchers must be cognizant that simple manipulation of the quarter to collect samples may lead to fluctuations in the SCC concentrations (as we observed in the untreated control in the present study), which may lead to confounded outcomes. A question that can be derived from this includes is the increase in SCC (or other features) on the control quarters caused by manipulation or an indirect effect of the infused quarter? To avoid any misleading information, we pursued the design with the current approach.

Materials and methods must be improved for example methods for evaluating the biochemical profile of Albumin, Globulin, GGT, and CK cholesterol must be better described including kits and equipment in order to increase the reproducibility of the work,

AU: The machine used is a Clinical Chemistry Analyzer AU680 (Beckman Coulter Inc, Brea, CA) that has now been added to the manuscript. The machine has specific standard methods for each analyte. This is a machine used routinely in Veterinary Diagnostic Laboratories and the approach used in the current has been used as a gold standard in several other publications. Please see the examples below. We are also attaching a document containing specifications and references for the metabolites evaluated by the machine as supplemental file, but we don’t think that is need to have it included in the manuscript to make the methodology used repeatable.

  • Journal of Dairy Science: DOI: 10.3168/jds.2018-15688

https://www.sciencedirect.com/science/article/pii/S0022030219306551

  • Lab. Med. DOI: 10.1016/j.plabm. 2019.e00131

https://www.sciencedirect.com/science/article/pii/S2352551719300502

  • Journal of Dairy Science: DOI: https://doi.org/10.3168/jds.2019-17527

https://www.sciencedirect.com/science/article/pii/S0022030220300102

  • J Vet Intern Med.: doi: 10.1111/jvim.13608

https://onlinelibrary.wiley.com/doi/full/10.1111/jvim.13608

Flow cytometry methods could be better described including results in section 2,1
AU:
A picture containing a representative plot of the flow cytometry quadrants (Figure 1) for each dye, was created to help to elucidate the results.

Section 2.2 authors could made figures for the results of the first and second paragraph
(AU):
Thank you for the suggestion. Considering that those results are not the main ones, we opted to keep those results only mentioned in the text, especially considering the number of pictures already posted in the manuscript.

The authors could include the mastitis history of the animals used in the present work, even to justify the non-comparison between mammary quarters of the same animal.

(AU): Thanks for the suggestion. We checked which of the animals had elevated SCC conditions without signs of clinical mastitis and a history of other concurrent diseases before purchasing and these were the criteria to enroll these cows in the study with high SCC in the study. The information has been added to the manuscript.

 The authors could describe how the number of animals to be tested was decided, even because the comparison of parameters is between animals and not intra-animal.
(AU):
  The power analysis is described on line 587. The mean difference expected between treatments, standard deviation, and reference is mentioned in that paragraph. The rationale for the use of only one quarter is explained in suggestion #1 of the present letter.

Round 3

Reviewer 2 Report

 The authors answer the questions satisfactorily